# Clinical Evidence on the Potential Beneficial Effects of Probiotics and Prebiotics in Cardiovascular Disease

**DOI:** 10.3390/ijms232415898

**Published:** 2022-12-14

**Authors:** Eleni Pavlidou, Aristeidis Fasoulas, Maria Mantzorou, Constantinos Giaginis

**Affiliations:** Department of Food Science and Nutrition, School of Environment, University of the Aegean, 81400 Myrina, Lemnos, Greece

**Keywords:** cardiovascular disease, clinical studies, gut microbiota, prebiotics, probiotics, coronary artery disease, stroke, hypertension

## Abstract

The ‘gut microbiome’—the hundreds of trillions of bacteria in the human gastrointestinal tract—serves several functions. The gut microbiome includes all the microorganisms, bacteria, viruses, protozoa, and fungi in the gastrointestinal tract and their genetic material. It helps digest indigestible foods and produces nutrients. Through the metabolism of sugars and proteins, it helps the intestinal barrier, the immune system, and metabolism. Some bacteria, such as those in the gut microbiome, cause disease, but others are essential to our health. These “good” microbes protect us from pathogens. Numerous studies have linked an unhealthy gut microbiome to obesity, insulin resistance, depression, and cardiometabolic risk factors. To maximize probiotic benefits in each case, knowledge of probiotic bacterial strains and how to consume them should be increased. This study aims to examine the benefits of probiotic and prebiotic organisms on cardiovascular health, specifically on heart disease, coronary heart disease, stroke, and hypertension. To complete the research, a literature review was conducted by gathering clinical studies and data. The clinical evidence demonstrates the beneficial effect of probiotics and prebiotic microorganisms on the gut microbiome, which has multiple benefits for overall health and especially for cardiovascular diseases.

## 1. Introduction

The gut microbiome contributes to food metabolism and the immune system, and is being strongly investigated as a diagnostic and therapeutic factor for various cardiovascular diseases. Evidence from research suggests that there is a strong correlation between the gut microbiome and the development of cardiovascular disease. In particular, a correlation has been found between the gut microbiome and the production of N-trimethylamine oxide, derived from dietary components such as choline and carnitine. Certain bacteria that are found in the gut are capable of converting choline and carnitine (components that are found in red meat and other animal foods) into N-trimethylamine oxide, potentially increasing risk factors for heart disease [1].

In patients with pre-existing cardiovascular disease, the microbial-induced conversion of ingested choline to trimethylamine affects the health of the cardiovascular system. Elevated blood levels of N-trimethylamine oxide have been directly linked to adverse outcomes in patients with such conditions, such as coronary artery disease and heart failure. Over time, the microbiome begins to produce toxic molecules, including N-trimethylamine oxide, which enter the bloodstream, causing inflammation. The age-related microbial imbalance of the gut microbiome contributes to the development of oxidative stress and inflammation that underlie arterial dysfunction. Such findings, therefore, support that good gut microbiota helps prevent cardiovascular disease [2].

As is now well understood, the gut microbiome provides many of the most important functions for our body. It is now recognized as a key component of human health and has been proposed to be defined as an “organ” of the human body, strange as that may sound. The microorganisms of the gut microbiome stimulate the immune system by breaking down toxic food compounds and synthesizing vitamins and amino acids, including vitamin B12 and vitamin K [3]. Also, a healthy microbiome full of “good” bacteria provides protection against pathogens entering the body. In addition, microbes that are found in the gut prevent the overgrowth of harmful bacteria that compete for certain nutrients [4].

The gut microbiota is a diverse symbiotic community of nonpathogenic microorganisms made mostly of anaerobic bacteria and fungi [5]; however, certain gut bacteria prefer to flourish under microaerophilic circumstances. One of its capabilities is the maintenance of a barrier via enterocytes that are covered with a brush border of mucus, which is produced by goblet cells and non-penetrable tight junctions between enterocytes [6], a layer of luminal mucus and tight adherent junctions between enterocytes that allows the control of absorption and metabolism as well as the maturation and stimulation of the immune system, all of which are essential functions for an effective mechanism of defense against pathogenesis. The signals that are generated by the bacteria in the gut travel through the intestinal epithelium and stimulate various signaling mechanisms at the epithelial cell boundary before entering the systemic circulation. This allows the signals to communicate with other organs that are located further away. The two-way connection that exists between the microbiota in the stomach and the organs throughout the body is mediated in healthy people by a few different pathways [7]. 

Therefore, if an appropriate entry route can be identified, probiotics can be a beneficial treatment. Multiple clinical investigations have proven the beneficial effects of probiotics in stroke patients. Due to the variable quality and sample sizes of these studies, it is not possible to evaluate the efficacy and safety of probiotic medication combined with EN in a systematic manner. Therefore, this article will explore further clinical evidence highlighting the protective use of prebiotics and probiotics in CVD, coronary heart disease, stroke and hypertension.

## 2. Reduction in the Risk Factors of CVDs

In a recent meta-analysis, Mo et al. [8] concluded that probiotics significantly reduce total cholesterol and low-density lipoprotein cholesterol in hypercholesterolemic individuals. 

Systemic inflammation is a prevalent feature of cardiovascular diseases (CVD), as are several risk factors such as hypertension [9]. It is also recognized that oxidative stress has a role in the progression of CVDs [10]. It has been discovered that Lactobacillus and Bifidobacterium inhibit lipid peroxidation and ROS production and, as a result, may slow or even prevent the development of CVDs and other oxidative stress-related diseases [11]. 

Tenorio-Jimenez et al. [12] report that the 12-week administration of a daily dose of 5 × 10^9^ cfu L. reuteri V3401 in capsules was associated with lower levels of inflammation biomarkers, such as TNF-, IL-6, IL-8, and soluble intercellular adhesion molecule-1 (sICAM-1), and a reduced risk of CVD in obese adults aged 18 to 65 years with metabolic syndrome. Szulinska et al. [13] also discovered that treatment with the multispecies probiotic Ecologic^®^ Barrier modified both the functional and biochemical markers of vascular dysfunction in 81 obese postmenopausal Caucasian women. The subjects were assigned to placebo, low dose (2.5 × 10^9^ cfu per day), or high dose groups (1 × 10^10^ cfu per day). The supplement was administered for 12 weeks. According to the researchers, both low and high doses reduce systolic blood pressure, serum vascular endothelial growth factor (VEGF), TNF-α, and IL-6. Other clinical studies [14,15] have similarly connected the use of probiotics to a moderate or substantial reduction in blood pressure in both healthy and obese individuals. It is believed that probiotics exert their antihypertensive effect through many mechanisms, including modulation of the renin–angiotensin system [16].

In the most recent study that was released by the World Health Organization, it was stated that strokes are the second biggest cause of death all over the world [17]. In 2016, a stroke was responsible for the deaths of nearly 5.5 million people [18]. The release of digestive hormones and neurotransmitters can be altered as a result of nerve damage that is caused by a stroke. This, in turn, causes the function of the intestinal mucosa to become disrupted, which makes digestion and the absorption of nutrients from the intestines more difficult. Due to this, the installation of a nasogastric tube for enteral feeding (EN) is necessary in order to guarantee that the body will receive adequate nutrition. At the moment, the bulk of nutritional support treatments involve either enteral feeding or parenteral nutrition (PN) [19]. However, delivery of PN [20] for extended period of time may result in undesirable effects, such as complications with the catheter and harm to the intestinal mucosa. In addition to this, EN is responsible for the maintenance of the gastrointestinal tract’s barrier function as well as an increase in the proliferation of intestinal mucosal cells. As a result, there is a relatively low occurrence of adverse reactions to EN. As a result of considerable gastrointestinal dysfunction, patients who have suffered a major stroke are more likely to experience difficulties such as diarrhea, constipation, and infection within one to two weeks of initiating EN therapy. This makes the application of EN [21] more difficult and reduces its effectiveness. Probiotics are live bacteria that are beneficial to the host and have the potential to colonize the human digestive tract. The human body can reap the benefits of having a microecology in the gut that is in good health. Wong [22] found that prompt probiotic supplementation can minimize intestinal permeability in critically ill patients, reduce the generation of pathogenic toxins and gas, lessen abdominal distension, neutralize food allergies, and lessen the severity of symptoms that are associated with irritable bowel syndrome, and increase EN tolerance.

## 3. Cardiovascular Disease

The term “cardiovascular disease” covers a wide range of diseases, including all pathological changes involving the heart and/or blood vessels [23]. These diseases include hypertension, coronary heart disease, heart failure, angina pectoris, myocardial infarction, and vascular strokes. Over half of deaths in the middle-aged and one-third of deaths in the elderly are attributed to CVD in the majority of developed nations [24]. The statistics related to cardiovascular disease appear ominous at the global level. Cardiovascular disease has been the leading cause of death in developing countries for the last 15 years, and by 2030, deaths will exceed 20 million per year [25]. 

The most common cause of heart disease is atherosclerosis (plaques in the artery walls which make them thicker, thus preventing proper blood flow). Other causes include high blood pressure, diabetes, coronary heart disease, stress, heart damage, bacteria, viruses, parasites (for heart infection), smoking, alcohol, and diet. Numerous studies have examined the beneficial effects of probiotics and prebiotics on host’s health. The term probiotic comes from the Greek, where it means “for life”. The definition of probiotics established in 2014 by the Food and Agriculture Organization of the United Nations (FAO) and the World Health Organization (WHO) is “live strains of strictly selected microorganisms that, when administered in adequate quantities, confer a health benefit on the host” [26]. Gibson and Roberfroid first proposed the term “prebiotics” in 1995 and updated it in 2004 as “non-digestible food components that allow the specificity of microbial changes in the intestinal tract, thereby exhibiting beneficial effects on the host’s health” [27]. 

Currently, the International Scientific Association for Probiotics and Prebiotics (ISAPP) reaffirms the most widely accepted definition of this term as a substrate that is selectively utilized by host microorganisms that confer a health benefit [26]. Probiotics and prebiotics have attracted a lot of interest in terms of the health benefits they may have. Although research is ongoing, it appears that probiotics may help treat diarrhea that is caused by taking antibiotics, treat urinary tract infections, treat irritable bowel syndrome (IBS), speed up the treatment of intestinal infections, and prevent or reduce the severity of colds and flu. Their intake is naturally through some types of food or more enhanced through supplements, as needed. Accumulating evidence suggests that probiotics and prebiotics may ameliorate metabolic disorders, such as obesity, diabetes, and CVD [28]. It has been demonstrated that they protect against CVD by lowering cholesterol levels, reducing oxidative stress, balancing functional and structural changes of gut microbiota, and enhancing immune responses [29]. 

Thanks to modern high-throughput techniques for sequencing gut microbiota, the role of gut microbiota in human health and well-being is the subject of extensive research [30]. Researchers suggest that heart patients may benefit from being treated with probiotics which colonize the gut with friendly bacteria. After all, heart disease is on the list of inflammatory problems such as rheumatoid arthritis, psoriasis, inflammatory bowel disease, and multiple sclerosis that also originate in the gut and have all been linked to a gut microbiota imbalance [31,32]. Probiotics and prebiotics with beneficial effects on the microbiological and metabolic makeup of gut microbiota could be considered as a potential therapy for CVD. 

Modulation of the host immune system could potentially account for the protective effects of probiotic and prebiotic treatments on cardiovascular disease. Immunological mechanisms behind probiotics and prebiotics involve dendritic cells, epithelial cells, T-regulatory cells, effector lymphocytes, natural killer T-cells, and B-cells [33]. Cardiovascular disease is associated with low-grade inflammation, as is the case with many chronic conditions. Plasma concentrations of the proinflammatory factors interleukin-1, interleukin-6, and tumor necrosis factor, which are stimulated by innate and adaptive immune cells, are commonly detected in cardiovascular diseases [34].

CVD refers to a collection of disorders with a complicated origin and pathogenesis. Raygan reported that supplementation with probiotics and co-supplementation with probiotics and vitamin D or selenium could significantly improve the biomarkers of mental health and metabolic profiles, such as high-sensitivity C-reactive protein, nitric oxide, low-density lipoprotein or total cholesterol, as well as the parameters that are involved in inflammation and oxidative stress [35,36]. Several indicators, including serum high-sensitivity C-reactive protein, cholesterol, plasma nitric oxide, and malondialdehyde (MDA), improved following supplementation with synbiotics including probiotic strains and prebiotic “inulin”, according to two further investigations [37]. The last three articles are associated with coronary artery disease (CAD). Supplementation with *L. plantarum* 299v (Lp299v) improved vascular endothelial function and reduced system inflammation in males with CAD [38]. *Lactobacillus rhamnosus* GG consumption was also associated with a reduction in metabolic endotoxemia and mega-inflammation in CAD patients [39]. In a further trial that was conducted by the same group, co-supplementation of probiotics and inulin with CAD patients for eight weeks was found to have favorable effects on biomarkers of depression, anxiety, and inflammation [40]. 

The positive benefits of soluble fiber (Minolest) supplementation on the lipid profile of people with mild hypercholesterolemia and a low risk of CAD were determined [41]. In a rat model of ischemia-reperfusion, another study demonstrated that larch arabinogalactan, an active component of pectin, reduced cardiac damage by blocking apoptotic cascades [42]. Moreover, chitosan oligosaccharides displayed preventive effects in CHD by enhancing antioxidant capabilities and lipid profiles by boosting the proliferation of probiotic species in the gut flora [43]. Overall, prebiotics may ameliorate CVD symptoms via many mechanisms involving inflammation, antioxidant capacity, and resetting the dysbiotic gut microbiome. However, the observed deleterious effects of prebiotics on CVD require caution in the human application of inulin.

## 4. Coronary Artery Disease

Numerous studies have found that dysbiosis contributes to the development of CAD through various mechanisms such as increased intestinal permeability and metabolic endotoxemia [44]. This could be explained by a microbiome-derived lipopolysaccharide, which is a primary component of Gram-negative bacteria’s exterior membrane. Lipopolysaccharides can enter the bloodstream after passing through the intestinal mucosa and may be a major modulator of chronic inflammation [45]. Although endotoxaemia is not always synonymous with high lipopolysaccharides, many people define metabolic endotoxemia as “a state of chronically elevated plasma lipopolysaccharides” [46]. Chronic inflammation after metabolic endotoxemia may be a plausible mechanism explaining the link between dysbiosis and CAD, which is caused by dysbiosis [47]. Lipopolysaccharides activate Toll-like receptors and causes endothelium damage by raising the expression of surface adhesion molecules such as a cluster of differentiation 14 on inflammatory cells, as well as stimulating the release of proinflammatory cytokines [48]. Endotoxin may also cause plaque development and progression of atherosclerotic lesions, as well as the release of other chemicals that are implicated in the proinflammatory process from endothelial cells [49].

Probiotics, in addition to aiding in the maintenance of homeostasis in the gut microbiota, have been proposed as a viable treatment for CAD [50]. In some therapeutic contexts, a small number of researchers have looked into the impact of probiotics on systemic levels of endotoxin. Those who have studied the effect of probiotics on endotoxin levels and associated metabolic diseases have found contradictory results [51]. Probiotics maintain gut barrier function integration and decrease intestinal permeability, lowering endotoxin levels [52].

## 5. Stroke

A stroke is an acute cerebrovascular disease that primarily manifests as a blockage of blood vessels in the brain. Ischemic or hemorrhagic strokes are the most common kinds of stroke, and males over the age of 40 are more likely to suffer from them. The two clinical treatments for stroke that are most commonly used are thrombolytic therapy and drug therapy. Due to the substantial risk of problems that are associated with both of these treatments, there is no improvement in the prognosis for stroke [53]. Nutritional support therapy is an important intervention that should be used in the treatment of acute severe stroke. To account for the treatment and the provision of appropriate nutrition for recovery in later stages, most studies have utilized early EN maintenance therapy [54]. Additionally, researchers have discovered that EN is appropriate for individuals with any awareness issue [55]. However, Xu discovered that EN can elicit a range of unpleasant gastrointestinal symptoms [56]. Probiotics can lower the incidence of complications and limit the growth of pathogenic bacteria in the intestine [57]. Using the notion of biological antagonism to modify the balance of gut flora is also consistent with contemporary concepts of medical treatment. 

Nevertheless, the link between probiotics and stroke is complex. Ritzel discovered that the incidence of intestinal dysbiosis in elderly stroke patients is on the rise [58]. Yin discovered an increase in harmful bacteria and a decrease in probiotics in the gut flora of ischemic stroke patients [59]. Recent studies [60] indicate that an imbalance in the gut flora can influence the incidence of stroke via a bottom-up signaling pathway. Consequently, gut inflammation and immunological response may be linked [61]. Probiotics and their metabolites, such as short-chain fatty acids, can dramatically ameliorate the systemic inflammatory response syndrome in critically ill patients [62]. 

## 6. Hypertension

Arterial hypertension is a key risk factor for the development of serious disorders, such as acute myocardial infarction, heart failure, stroke, and renal failure [63], in addition to being a leading cause of early death around the world [64]. Primary hypertension, also known as essential hypertension, is a complicated illness that can be caused by several variables, including genetics, demography, concurrent disorders, and the environment. Approximately 8% of cases are suggestive of secondary hypertension, or hypertension with a known etiology including endocrine disorders, medications, malignant tumors, or hyperactivation of the renin-angiotensin system [65]. 

Antihypertensive medication used in clinical practice has been shown to be effective at keeping blood pressure at safe levels, hence lowering the morbidity and mortality that is linked to this condition. Several worldwide publications, the bulk of which were released by the Eighth Joint National Committee, have created guidelines for the treatment of hypertension patients as well as numerous reference values for those over 60 [66]. These guidelines were developed by several international publications. 

In addition to pharmacological therapies, it is important to implement several nonpharmacological methods for the disease’s control. In this regard, a healthy diet is of critical importance [67], and probiotics can play a crucial role. Several mechanisms, such as exerting control at the level of the central and autonomic nervous systems or safeguarding endothelium function, have indicated that gut microbiota plays a significant role in the regulation of blood pressure. In addition, gut dysbiosis has been described in hypertensive animal models and hypertensive people [68]. 

Considering the use of probiotics, a working committee at the National Heart, Lung, and Blood Institute recently examined the current status and future directions for the treatment and prevention of high blood pressure, taking into account the role of the gut flora. Therefore, functional foods that contribute to the preservation of the intestinal flora can be highly helpful in preventing excessively high blood pressure levels, as various writers have already discussed in detail [69]. 

Kefir is one of the foods that has been proven to deliver numerous cardiovascular advantages, and one of those advantages is an effective reduction in blood pressure [70]. Kefir has been studied for its ability to inhibit the angiotensin-converting enzyme, as well as for the prevention of vascular endothelial dysfunction [71], the restoration of damaged autonomic cardiovascular function [72], and its inhibition of angiotensin-converting enzyme [73]. Mounting data are suggesting that the use of probiotics as a viable natural coadjuvant in the treatment and prevention of cardiovascular disease, including hypertension, may be possible.

Multiple human clinical investigations have revealed that probiotics can reduce unusually high blood pressure levels. For instance, systolic/diastolic blood pressure and heart rate decreased in response to an extract of *Lactobacillus casei*, which has been shown to lower blood pressure in spontaneously hypertensive patients [74]. An intriguing 2002 study found that using *Lactobacillus plantarum* lowered heavy smokers’ systolic blood pressure [75]. Chronic probiotic use lowers the risk of preeclampsia, which is linked to inflammation and hypertension according to a 2011 Norwegian study [76]. Additionally, probiotic soy milk containing *Lactobacillus plantarum* significantly decreased systolic/diastolic blood pressure in a randomized, double-blind clinical trial with Type II diabetes mellitus [77], and in a study with pre-diabetic patients, there was a significant tendency to reduce hypertension in those patients receiving a multispecies probiotic [78]. Consuming probiotics resulted in a little reduction in blood pressure, with the impact being more pronounced if the basal blood pressure was elevated, according to a meta-analysis that was published in 2014 based on the results of nine clinical trials. The authors also concluded that combining different probiotic strains increased their effectiveness. Last but not least, the intervention must last no more than eight weeks, and the daily probiotic dose cannot exceed 1011 colony-forming units [79]. 

The initial line of defense between the bloodstream and the vascular muscle is made up of a single layer of thin, smooth cells known as the vascular endothelium. Its duties include acting as a selective membrane for the movement of fluid, solutes, and inflammatory cells between the tissue and plasma areas [80]. 

The endothelium also modulates vascular tone by generating and releasing a variety of vasodilators and vasoconstrictors, including nitric oxide (NO), prostacyclin, and endothelium-derived hyperpolarizing factor and endothelin (through ETA), angiotensin II (by AT1 receptors), and ROS. Furthermore, it regulates platelet aggregation and blood hemostasis, as well as the antithrombotic/prothrombotic balance, and it plays a function in inflammation and immunological response [81].

Due to the endothelium’s multifunctional character, it is easy to see how changes in it can play a role in the genesis and/or advancement of numerous diseases. Endothelial dysfunction is thus recognized as a risk factor for the onset of CVD, manifesting itself in the early stages and during the progression of hypertension, myocardial ischemia, atherosclerosis, and peripheral vascular disease [82]. Endothelial dysfunction is also a component of disorders such as diabetes, renal failure, viral infections, and tumor growth [83]. 

Inflammatory processes can cause endothelial dysfunction, which leads to a decrease in endothelial nitric oxide synthase (eNOS) enzyme activity, a decrease in nitric oxide bioavailability, and hypertension. Furthermore, oxidative stress promotes the development of endothelial dysfunction by reducing nitric oxide availability [84]. Endothelial nitric oxide release is reduced when ROS are produced as a result of hypertension, hypercholesterolemia, diabetes, or other cardiovascular risk factors [85].

As stated previously, there is a significant association between dysbiosis and the development of hypertension, which may include the decreased endothelium function that is caused by changes in the gut microbiota during the chronic rise in blood pressure [86]. Consequently, a number of studies have demonstrated that probiotics may improve endothelium function [87].

Some research on humans or human cells has demonstrated that probiotic therapy improves endothelial function. Soy milk fermented with *Lactobacillus plantarum* or *Streptococcus thermophilus* enhanced nitric oxide production and eNOS activity in endothelial cells, suggesting their efficacy for enhancing endothelial function [88]. In men with stable CAD, a 6-week supplementation with *Lactobacillus plantarum* improved endothelial function for both conduit and resistance vessels by increasing nitric oxide bioavailability and concurrently reducing systemic inflammation, as measured by brachial artery flow-mediated dilation. These findings indicate that the gut microbiota is mechanistically linked to systemic inflammation and endothelial function [89]. Another clinical trial demonstrated that a multispecies probiotic supplement improved both functional and biochemical parameters of endothelial dysfunction in obese postmenopausal women. These parameters included systolic blood pressure, vascular endothelial growth factor, pulse wave velocity and its augmentation index, interleukin-6, TNF-α, and thrombomodulin. In contrast, in a study of people with metabolic syndrome who received *Lactobacillus casei* Shirota supplementation, no significant changes in markers that were used to measure low-grade inflammation or endothelial dysfunction were observed [90]. 

In general, in vivo and in vitro investigations, as well as clinical studies in people, indicate that treatment with numerous types of probiotics improves endothelium function via multiple routes. Despite the need for additional investigation, the effect of probiotic supplementation in preventing CVD by addressing endothelial dysfunction appears promising. In addition, the multifunctional nature of the endothelium broadens the potential application of probiotics to all diseases, not just cardiovascular, whose pathogenesis may be associated with endothelial dysfunction.

## 7. Atherosclerosis

The primary risk factor for CVD is atherosclerosis, which is characterized by cholesterol buildup and the migration of macrophages into arterial walls, which helps produce atherosclerotic plaques [91]. It is interesting to note that recent research has revealed that gut dysbiosis may possibly be a factor in the onset of atherosclerosis [92]. Researchers discovered that the relative abundance of Roseburia and Eubacterium was lower, whereas Collinsella was higher, in atherosclerosis patients compared to healthy controls, using shotgun sequencing of the gut metagenome in patients with or without symptomatic atherosclerosis [93]. The host’s first line of defense against the invasion of pathogens is the gut epithelium [94]. The integrity of the gut barrier is crucial for sustaining the host’s health due to its crucial role in preventing the transfer of intestinal material, primarily bacterial components. Reduced expression of tight junction proteins such as zonula occludens-1 (ZO-1), claudin-1, and occludin, as well as an imbalance between intestinal epithelial cell death and regeneration, are all related with intestinal permeability [95]. When the intestinal epithelial barrier is compromised, pathogen-associated molecular patterns (PAMPs) invade and trigger an immune response that leads to both systemic and tissue-specific inflammation. As such, it has been proposed that gut dysbiosis, which impairs the integrity of the gut barrier, is a risk factor for chronic inflammation in a number of disorders. It is interesting that peptidoglycan and lipopolysaccharides are microorganisms that have been linked to CVD risk.

## 8. Conclusions

The positive effects of probiotics and prebiotics on host health have been the subject of numerous studies. Probiotics appear to have potential as a treatment for a variety of conditions, including irritable bowel syndrome, urinary tract infections, cold and flu symptoms. Research into this topic is still ongoing. When necessary, supplements are used to supplement their natural intake through some food types. Probiotics and prebiotics may help treat metabolic disorders such as obesity, diabetes, and cardiovascular disease, according to growing evidence. They have been shown to defend against CVD by lowering cholesterol levels, cutting down on oxidative stress, balancing the functional and structural alterations of the gut microbiota, and boosting immune responses. According to numerous studies, dysbiosis affects the development of CAD via several different mechanisms, including increased intestinal permeability and metabolic endotoxemia. Probiotics have been suggested as a potential treatment for CAD in addition to helping to maintain homeostasis in the gut microbiota. Probiotics’ effects on systemic levels of endotoxin have been studied in a small number of therapeutic contexts. Researchers who have looked into how probiotics affect endotoxin levels and related metabolic diseases have come up with conflicting findings. Strokes and probiotics have a complicated relationship. According to recent research, a bottom-up signaling pathway may be used to affect the incidence of stroke by affecting the gut flora. Increasing evidence points to the possibility of using probiotics as an effective natural coadjuvant in the treatment and prevention of cardiovascular disease, including hypertension.

Evidence has been presented in the scientific literature indicating that a regular intake of probiotics, which restore the balance of the intestinal microbiota, may provide cardiovascular benefits based at least in part on its ability to reduce oxidative stress. Although many questions remain unanswered and many published results are contradictory, it is clear that the consumption of probiotics represents a promising complement to more conventional cardiovascular therapies, as well as to nonpharmacological measures that are commonly used to prevent the onset and progression of CVD. Further research is necessary to clarify the interaction between the gut microbiota, the neuroimmune system, and the endocrine system in order to develop nutrigenetic profiles that may aid in achieving homeostasis. In addition, it will be necessary to increase knowledge regarding the various bacterial strains that are present in probiotics and how they should be consumed to maximize their potential beneficial effects in each particular circumstance. Lastly, studies of the vast variety of enzymes, peptides, and biochemical pathways that are produced by the intestinal microbiota, which differ from the host’s resources, could serve as an innovative strategy for the design of new drugs to treat CVD.

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
