# Peer review of "Clinical Evidence on the Potential Beneficial Effects of Probiotics and Prebiotics in Cardiovascular Disease"

_ijms, 2022, doi:10.3390/ijms232415898_

Round 1

Reviewer 1 Report

Please go thoroughly throughout the guide for authors of International Journal of Molecular Sciences and made necessary corrections to your manuscript and please correct the grammar – same are marked directly in the text.

Line 16: The aim of 16 this study is to examine  - I propose: This study aims to examine….

Line 46: that a good – correct that good….

Line 71: S. thermophilis, L. acidophilus LA-5 and B. bifidum – correct S. thermophilis, L. acidophilus LA-5 and B. bifidum - Latin names of bacteria in italics

Line 86: study studied  - I propose: study examined

Line 125: …for the maintenance of the barrier function of the gastrointestinal tract... – I propose: ….for the maintenance of the gastrointestinal tract's barrier function.... 

Line 155: …which makes... – correct ...which make...

Line 157: parasites (for heart infection) smoking, alcoholand diet. – correct: parasites (for heart infection), smoking, alcohol and diet.

Line 158: host health – correct host's health

Line 193: is associated by low-grade – correct is associated with low-grade

Line 231: endotoxaemia - elsewhere in the text is endotoxemia?

Line 237: such cluste – correct: such as a cluste

Line 243: research – correct: researchers

Line 256: treatment – correct: the treatment

Line 275: therapy – correct therapeutic

Line 303: a number of – I propose several

Line 307: the gut microbiota – correct: gut microbiota

Line 321: to inhibit angiotensin – correct: to inhibit the angiotensin

Line 324: There is mounting data suggesting – I propose: Mounting data is suggesting

Line 374: transplantation – correct: the transplantation

Line 385: analogues – I propose analogs

Line 394: have – correct has

Line 412: multiple – I propose numerous

Line 421: diarrhoea – correct diarrhea

Line 429: a number of – I proprose several

Line 441: have – correct has

Author Response

Reviewer 1

We would like to thank the reviewer for their useful comments and suggestions

Please go thoroughly throughout the guide for authors of International Journal of Molecular Sciences and made necessary corrections to your manuscript and please correct the grammar – same are marked directly in the text.

Line 16: The aim of 16 this study is to examine - I propose: This study aims to examine….

It has been corrected

Line 46: that a good – correct that good….

It has been corrected

Line 71: S. thermophilis, L. acidophilus LA-5 and B. bifidum – correct S. thermophilis, L. acidophilus LA-5 and B. bifidum - Latin names of bacteria in italics

It has been corrected

Line 86: study studied  - I propose: study examined

It has been corrected

Line 125: …for the maintenance of the barrier function of the gastrointestinal tract... – I propose: ….for the maintenance of the gastrointestinal tract's barrier function.... 

It has been corrected

Line 155: …which makes... – correct ...which make...

It has been corrected

Line 157: smoking, alcoholand diet. – correct: parasites (for heart infection), smoking, alcohol and diet.

It has been corrected

Line 158: host health – correct host's health

It has been corrected

Line 193: is associated by low-grade – correct is associated with low-grade

It has been corrected

Line 231: endotoxaemia - elsewhere in the text is endotoxemia?

It has been corrected

Line 237: such cluste – correct: such as a cluste

It has been corrected

Line 243: research – correct: researchers

It has been corrected

Line 256: treatment – correct: the treatment

It has been corrected

Line 275: therapy – correct therapeutic

It has been corrected

Line 303: a number of – I propose several

It has been corrected

Line 307: the gut microbiota – correct: gut microbiota

It has been corrected

Line 321: to inhibit angiotensin – correct: to inhibit the angiotensin

It has been corrected

Line 324: There is mounting data suggesting – I propose: Mounting data is suggesting

It has been corrected

Line 374: transplantation – correct: the transplantation

It has been corrected

Line 385: analogues – I propose analogs

It has been corrected

Line 394: have – correct has

It has been corrected

Line 412: multiple – I propose numerous

It has been corrected

Line 421: diarrhoea – correct diarrhea

It has been corrected

Line 429: a number of – I proprose several

It has been corrected

Line 441: have – correct has

It has been corrected

Reviewer 2 Report

This review summarized the potential beneficial effects of probiotics and prebiotics in cardiovascular diseases. The topic is interesting and the paper is well structured. The following issues should be tackled with before the consideration for acceptance.

1. The evidences on the anti-inflammatory, lipids-lowering and anti-oxidant effects of probiotics and prebiotics in CVDs are suggested to include solely in a part as reduction in the risk factors of CVDs, instead of in the introduction part.

2. Atherosclerosis is a popular target with intense investigation. Is there any clinical studies regarding the effects of probiotics and prebiotics in atherosclerosis?

3. The authors said the aim was to focus on clinical studies. However, a number of animal studies were included. I recommend to remove them in order to keep the focus.

4. Please check the paper carefully to avoid some minor spelling or language mistakes. For example, in line 123, there was an repeated word for.

Author Response

Reviewer 2

We would like to thank the reviewer for their useful comments and suggestionss

  1. The evidences on the anti-inflammatory, lipids-lowering and anti-oxidant effects of probiotics and prebiotics in CVDs are suggested to include solely in a part as “reduction in the risk factors of CVDs”, instead of in the introduction part.

“Reduction in the risk factors of CVDs” has been included as a separate subsection

  1. Atherosclerosis is a popular target with intense investigation. Is there any clinical studies regarding the effects of probiotics and prebiotics in atherosclerosis?

“Atherosclerosis” has been included as a separate subsection

  1. The authors said the aim was to focus on clinical studies. However, a number of animal studies were included. I recommend to remove them in order to keep the focus.

They have been removed

  1. Please check the paper carefully to avoid some minor spelling or language mistakes. For example, in line 123, there was a repeated word “for”.

The paper has been carefully checked.